# The Role of Corticosteroid Nasal Irrigations in the Management of Chronic Rhinosinusitis: A State-of-the-Art Systematic Review

**DOI:** 10.3390/jcm12103605

**Published:** 2023-05-22

**Authors:** Christian Calvo-Henriquez, Jaime Viera-Artiles, Miguel Rodriguez-Iglesias, Paula Rodriguez-Rivas, Antonino Maniaci, Miguel Mayo Yáñez, Gabriel Martínez-Capoccioni, Isam Alobid

**Affiliations:** 1Rhinology Study Group of the Young-Otolaryngologists of the International Federations of Oto-Rhino-Laryngological Societies (YO-IFOS), 70123 Paris, France; christian.ezequiel.calvo.henriquez@sergas.es (C.C.-H.); miguel94illa@gmail.com (M.R.-I.); paula.rodriguez.rivas2@sergas.es (P.R.-R.);; 2Service of Otolaryngology, Hospital Complex of Santiago de Compostela, 15701 Santiago de Compostela, Spain; 3Service of Otolaryngology, Rhinology and Skull Base Department, Marqués de Valdecilla Hospital, PC 39008 Santander, Spain; 4Department of Medical and Surgical Sciences and Advanced Technologies “GF Ingrassia” ENT Section, University of Catania, 95123 Catania, Italy; 5Service of Otolaryngology, Hospital Complex of La Coruña, PC 15001 La Coruña, Spain; 6Service of Otolaryngology, Rhinology and Skull Base Department, Clinic Hospital, PC 08036 Barcelona, Spain

**Keywords:** nasal rinse, budesonide, chronic rhinosinusitis, CRSwNP, CRSsNP, CRS, endoscopic sinus surgery, FESS, nasal polyps

## Abstract

Chronic rhinosinusitis (CRS) is a highly prevalent condition. CRS is usually managed with intranasal corticosteroids, useful both before as well as after endoscopic sinus surgery (ESS). However, the greatest drawback of these low-volume sprays is the inadequate delivery into the paranasal sinuses, even after ESS. Recent studies have shown that high-volume steroid nasal rinse (HSNR) has a significantly better penetration of the paranasal sinuses. The purpose of this state-of-the-art review is to systematically overview the current literature about the role of nasal rinses with steroids in CRS. Four authors examined four databases (Embase, Pubmed, Scielo, Cochrane). This review identified 23 studies answering 5 research questions. It included 1182 participants, 722 cases, and 460 controls. Available evidence suggests a potential positive effect of HSNR, which seems to be higher in CRS with nasal polyps. More well-designed studies are needed in order to obtain solid conclusions. The evidence is solid regarding the safety of this treatment modality in the short and long-term. We expect that this lack of severe negative effects will facilitate the acceptance of this treatment modality and the development of future studies.

## 1. Introduction

Chronic rhinosinusitis (CRS) is a highly prevalent condition, affecting approximately 7% to 14% of the North American population [1]. Its impact is significant, as it is linked to reduced quality of life (QoL) [2], impaired sleep [3], and fatigue [4].

CRS encompasses a variety of conditions and can be further categorized according to its phenotype into cases which present nasal polyps (CRSwNP) and cases which do not (CRSsNP) [3], each one being managed differently.

Both manifestations are usually treated with intranasal corticosteroids (INCS), which exert an anti-inflammatory effect by reducing airway inflammatory cell infiltration by eosinophils, mast cells, and T-lymphocytes, and suppressing production of adhesion molecules and pro-inflammatory genes and mediators, such as NF-κB [3].

When pharmaceutical intervention fails, endoscopic sinus surgery (ESS) is the preconized option [5]. One of the main targets and benefits of this surgery is the widening of the natural ostia in the sinus cavities, which enables an adequate distribution of the medication [6]. The proper distribution of the drug to the inflamed mucosa has been related to an improved clinical response [7]. Although the safety profiles for intranasal steroid sprays are well known, the greatest drawback of these low-volume aerosols is the inadequate delivery into the paranasal sinuses even after ESS.

Recent studies have shown that high-volume irrigations achieve a significantly better penetration of the paranasal sinuses, predominantly in the post-ESS cavity, compared to other delivery methods [8]. It has been proven that high-volume (>50 mL) and low-pressure nasal irrigation is the best method to reach all the nasal cavities and sinuses [9]. Consequently, high-volume steroid nasal rinses (HSNR) are believed to offer better control of the mucosal inflammation and have become very popular in recent years. Nevertheless, they are currently used off-label and are not FDA approved.

The effect of nasal rinses with steroids has been reported in CRS with and without nasal polyps and both before and after ESS. However, the available evidence draws from anecdotal experience and uncontrolled trials [10]. The purpose of this state-of-the-art review is to systematically overview the currently available literature about the role of nasal rinses with steroids in CRS, including both subjective and objective changes and side effects in order to guide daily practice, as well as to identify knowledge gaps so that future research can be improved, and further advances made.

## 2. Materials and Methods

This review was performed according to PRISMA guidelines [11], and a formal PROSPERO protocol was published according to the NHS International Prospective Register of Systematic Review (N° 399,100) prior to the initiation of the study. We also followed the recommendations of the AMSTAR-2 guidelines.

### 2.1. Literature Search: Inclusion and Exclusion Criteria

The criteria for considering studies for this review were based on the population, intervention, comparison, and outcome (PICOTS) framework.

**Participants**: Adults (>18 y.o.) suffering from chronic rhinosinusitis according to EPOS guidelines.

**Intervention**: Any type of steroid diluted in high-volume (>50 mL) nasal rinse.

**Comparison**: Before and after nasal lavage treatment from uncontrolled studies (quasi experimental studies), or intervention and non-intervention cohorts from controlled studies (cohorts and clinical trials).

**Outcomes**: Five research questions were defined: (1) Is high-volume steroid-diluted nasal rinse (HSNR) therapy useful in CRS before ESS? (2) Is HSNR therapy useful in CRS after ESS? (3) Which steroid is optimal? (4) Is HSNR better than intranasal steroid spray in the management of CRS? (5) Is HSNR safe in the short term (<3 months)? (6) Is steroid diluted high-volume nasal rinse therapy safe in the long term (>3 months)?

**Timing and Setting**: Without limitation.

**Types of studies**: Clinical trials, case series, and prospective and retrospective cohort studies published in peer-reviewed journals. Case reports, theses, narrative reviews, and meetings′ communications were not included. There were no restrictions by date or publication type. The search was last updated in February 2023. Studies were considered if published in English, Spanish, German, French, Italian and Portuguese.

**Exclusion criteria**: exclusion criteria consisted of (1) studies not individually assessing the effect of steroid nasal irrigation by including other CRS treatments such as monoclonal antibodies or ESS; (2) mixing different steroid nasal irrigation protocols without subgroup analysis; (3) studies not evaluating quantifying variables.

### 2.2. Search Strategy

We followed the recommendations of the PRISMA statement [11] to perform a systematic review and searched the following databases: Pubmed (Medline), the Cochrane Library, EMBASE, and SciELO. We used a predefined search strategy (polyp* [Title/Abstract] OR CRSwNP [Title/Abstract] OR sinusitis [Title/Abstract]) AND (Rinse [Title/Abstract] OR Irrigation* [Title/Abstract] OR Lavage [Title/Abstract] OR Douche [Title/Abstract]) AND (Steroid [Title/Abstract] OR Budesonide [Title/Abstract] OR Corticoid [Title/Abstract] OR corticosteroid [Title/Abstract]). The abstracts of the retrieved papers were thoroughly reviewed by four authors, members of the YO-IFOS rhinology study group (CCH, JVA, PRR, and MRI), and those potentially fulfilling the inclusion criteria were selected for full-text review. In case of discrepancies among reviewers regarding the selection of the abstracts, the relevant papers were included in the full text review stage for a final evaluation. The references of all selected articles have also been manually reviewed so as to identify any potentially missing publications.

#### Study Extraction, Categorization, and Analysis

Four authors (CCH, JVA, PRR, and MRI) independently analyzed the articles that met the inclusion criteria and extracted the relevant data. Discrepancies were resolved by discussion. Extracted variables encompassed sample size, inclusion criteria, exclusion criteria, age, sex, method of nasal rinse, type of nasal steroid, follow-up, studied variables before and after treatment, and main outcome.

### 2.3. Assessment of Study Quality

The selected articles were categorized according to both their level of evidence and quality. Level of evidence was classified according to the Oxford Centre for Evidence-Based Medicine Levels. The risk of bias was assessed according to the quality assessment of case series studies checklist from the National Institute for Health and Clinical Excellence (Appendix F) for quasi-experimental and case series studies (see Appendix C for randomized clinical trials).

### 2.4. Statistical Analysis

Data were analyzed with STATA for Macintosh v. 15.1 (StataCorp^®^, College Station, TX, USA) for means calculations (age, follow-up). Regarding included studies, statistical significance level was considered with a *p*-value < 0.05.

## 3. Results

### 3.1. Search Results

The PRISMA flowchart is presented Figure 1. The initial search retrieved 189 publications. After all titles and abstracts were read, 34 studies were selected for full text review. A total of 23 studies met the inclusion criteria.

One author was contacted with the objective of obtaining missing data [12].

Of the papers selected for full-text reading, 11 publications were excluded (references in Appendix A).

### 3.2. Results of the Included Studies

The search strategy yielded one meta-analysis; two systematic reviews, one narrative review; six randomized clinical trials, nine quasi-experimental studies; two prospective cohort studies; two retrospective; and two cross-sectional studies.

The whole retrieved sample (excluding reviews) included 1182, 722 cases and 460 controls. Its distribution was clearly asymmetric, as a single study [13] provided 50.1% of the whole included sample. The mean sample size was 62.2 while the median was 30, 9 being the minimum sample size [14] and 592 the maximum [13].

The mean age pondered by sample size and excluding reviews was 50.2 years; 51.9 for cases and 48.1 for controls.

Appendix A summarizes the evidence of each selected study. Table 1 summarizes the available evidence. Table 2 and Table 3 summarize the quality of the included papers.


**
*
Question 1: is HSNR therapy useful in CRS before ESS?
*
**


It is known that even in individuals not subjected to ESS, nasal irrigation has greater sinus penetration than nasal sprays [32], although less than in operated sinuses [33]. Therefore, the question arises of whether HSNR are useful in non-operated CRS patients. The role of previous surgery is discussed further on in this manuscript.


**
*
1a: Mixed CRS (CRSwNP and CRSsNP)
*
**


The effect of HSNR in mixed CRS before ESS was not evaluated by any of the selected papers.

The Luz-Matsumoto et al. [13] paper could not formally be included in this section as they mixed patients treated with different steroid therapies. However, it is noteworthy that they performed a subgroup analysis according to patients’ ESS status. Individuals who had previously undergone ESS exhibited a higher decrease in nasal symptoms.

To date, the available evidence is too scarce to draw any conclusions on the use of HSNR before ESS for CRS patients.


**
*
1b: CRSwNP
*
**


The effect of HSNR in CRSwNP before ESS was not assessed by any of the selected papers.


**
*
1c: CRSsNP
*
**


The effect of preoperative steroid nasal rinse in CRSsNP before ESS was assessed in one study [34].

Jiramongkolchai et al. [15] found statistically significant differences before and after treatment favoring the experimental group in the SNOT-22 and Lund–Kennedy scores in their double blind RCT comparing mometasone lavage against saline rinse with mometasone spray.


**
*
Question 2: is HSNR therapy useful in CRS after ESS?
*
**



**
*
2a: CRS (with and without NP)
*
**


The effect of HSNR in mixed CRS after ESS was assessed in six studies [6,16,17,18,19,20] including one systematic review and a meta-analysis [19].

Yoon et al reported a significant mean decrease in Lund–Kennedy and SNOT-22 scores—4.23 (95% CI: 3.60–4.86) and 21.92 (95%CI:13.95–29.89), respectively, in their meta-analysis. However, when data were compared against controls, there were no differences in the either the Lund–Kennedy score or in SNOT-22.

All of the other studies reported significant differences in clinical variables, with the exception of Harvey et al. [20]. In their RCT, they studied CRS patients undergoing ESS either receiving mometasone spray and placebo rinses or mometasone rinses with placebo spray. Compared against these controls, they could not identify any significant difference in the experimental group.

In conclusion, higher quality evidence is required to address this question. The available evidence suggests that HSNR may not be useful in mixed CRS after ESS compared to controls.


**
*
2b: CRSwNP
*
**


Three studies explored this question [21,22,23]. One quasi-experimental study [21] did find differences, while two RCT [22,23] could not identify any significant difference in relation to controls. Regarding Rotenberg et al.’s research [23], the question is whether this could be attributed to the severity of the disease (AERD) or, on the contrary, to the lack of efficacy of HSNR in CRSwNP. Regarding Rawal et al’s work, this lack of significance could be attributed to the fact that patients had recently undergone ESS and the follow-up was short (6 months). Relapse is not expected to occur that shortly. Therefore, this cohort of patients is not the most suitable in order to detect differences.

Given the quality of the evidence, and the contradictory results, no conclusions can be drawn.


**
*
2c: CRSsNP
*
**


The effect of SNR in CRSsNP after ESS was assessed in one RCT [24]. Thamboo et al. [24] compared budesonide when administered through an atomization device and through high-volume nasal rinse. They could not identify any SSD in the SNOT-22 score between groups. However, they only included 10 patients per arm treatment and the follow up was just 60 days.

The quality of the evidence is too low in order to obtain any conclusion.


**
*Question 3: which treatment modality is better?*
**


This review identified different treatment modalities with regard to variations in the type of steroid, irrigation volume and duration of the treatment. 

Regarding the type of steroid, 15 studies used budesonide [6,13,14,16,18,21,22,23,24,25,27,28,29,30,31], 2 mometasone [15,20], 1 fluticasone [26], and 1 bethametasone [13,17].

The comparison between different steroids is relevant. Lipophilic compounds are retained in the sinonasal tissue for longer periods of time, resulting in increased exposure to glucocorticoid receptors [35]. First generation steroids such as budesonide have an increased systemic bioavailability (30–60%), while second generation ones, such as mometasone or fluticasone, have less than 1%. However, only Luz-Matsumoto et al. [13] have compared different steroids, i.e., budesonide against betamethasone, with better results for budesonide.

In relation to dosage, there was also heterogeneity among the selected studies (Appendix A). Only Luz-Matsumoto et al. [13] compared different cohorts being administered different doses of steroids. The higher dose cohort (1000 mg/day) showed better results than the lower dose cohort (500 mg/day), with improvements in SNOT-22 (30.7–27.0; *p* = 0.06) and LKES (5.5–4.8; *p* = 0.03). 

In relation to the length of the treatment, there was some disparity, with periods of administration ranging from 30 days [14,16] up to 38 months [29]. None of the studies compared different cohorts according to length of treatment. 


*
Irrigation volume:
*


The selected studies ranged in the lavage volume from 120 mL to 240 mL, the last figure being the volume most used. According to previous studies, nasal lavage is effective from 100 mL [34]. However, it has been shown that the use of a 240 mL nasal irrigation bottle is the most effective in delivering medication to the postsurgical sinus cavity [36]. None of the selected authors compared different modalities in relation to steroid nasal rinse. 

Four studies were performed with volumes lower than 200 mL [14,18,22,23]. This could impair the distribution of the steroid into the nasal cavity and sinuses. It is noteworthy that two [22,23] of the three studies [13,22,23] with low-volume nasal rinse did not find differences in reported nasal symptoms (SNOT-22).


*
Treatment Duration:
*


This is another potential confounding factor in these clinical trials. In the two studies [16,22] that showed no statistically significant improvement in patients’ symptoms using nasal steroid irrigation, patients had been treated for fewer than 6 months. 

It is noteworthy that Kang et al. [21] reported maximal improvement at 2 months in a treatment which lasted for at least 6 months. However, given the wide disparity in treatment duration among the selected studies, the optimal duration for nasal steroid irrigation use remains unclear.


**
*
Question 4: is HSNR therapy better than intranasal steroid spray?
*
**


Six studies evaluated this postulate [10,13,15,18,20,23]. Grayson and Harvey [10] only found one study for their systematic review. Among the six, the work of Luz-Matsumoto et al. [13] stands out given its sample size. In their cohort trial, they compared steroid nasal spray vs. nasal rinses with betamethasone cream or budesonide. With their data, we performed a *t*-test which demonstrated a significant difference between the use of budesonide and intranasal steroids for the Lund–Kennedy score (*p* < 0.001).

Although more studies are needed, HSNR seems to be more effective than INCS in CRS patients after ESS, and also without prior ESS.


**
*
Question 5: which patients benefit the most from HSNR?
*
**


The unrestricted use of this treatment modality in all CRS patients may not be cost-effective, as it involves higher doses of steroids with more adverse events than INCS and with a greater chance of non-adherence due to the more complex preparation [37]. Hence, the importance of careful patient selection.

Kosugi et al. [6] divided their sample into respondents and non-respondents. They could not identify any predictor of response.

Jang et al. [18] performed subgroup analysis according to the subtype of CRS [allergic fungal rhinosinusitis; AERD; eosinophilic CRS with and without polyps) but did not compare the results among subgroups. They found worse results in AERD patients and no effect on allergic fungal sinusitis.


*
IgE and Eosinophils:
*


Tissue eosinophilia typically results in higher rates of postsurgical disease recurrence [38].

The response to steroid therapy could be different regarding the inflammation charge. There has been no subgroup analysis regarding the IgE, eosinophils count, or C-reactive protein. 


*
AERD:
*


Patients with AERD have a more severe disease phenotype and a higher recurrence rate after surgery.

Snidvongs et al.’s [12] work, not included in this review as they mixed different treatment protocols, performed subgroup analysis according to AERD diagnoses. They found statistically significant differences in both groups with NPS and SNOT-22, with a lower decrease in AERD patients.

Jang et al. [18] also performed subgroup analysis in AERD, showing a lower reduction in the SNOT-20 and Lund–Kennedy scores compared to the overall sample. Interestingly, Talat et al. [31] and Rotenberg et al. [23] performed their study only in CRSwNP patients with AERD. Rotenberg et al. [23] could not identify any differences between the use of HSNR and saline rinses alone. Talat et al. [31] found a reduction in SNOT-22 and nasal polyps score, but the study was not placebo controlled. 

Other authors reported this outcome. However, they had not performed subgroup analysis [20,29].


**
*
Question 6: is HSNR therapy safe in the short term (<3 months)?
*
**


Oral steroids have known side effects including suppression of the hypothalamic pituitary axis, osteoporosis, growth retardation, cataracts, and glaucoma. While intranasal spray steroids have proven to be safe, the safety of steroid rinses is not clear.

Given that a higher dose of steroids is administered through steroid nasal rinse rather than INCS, there has been concern regarding the safety profile of this practice in terms of systemic absorption, hypothalamic pituitary adrenal axis (HPAA) suppression, and elevated intraocular pressure (IOP). However, despite the higher quantity, less than 5% of the total rinse remains in the nose [33]. Therefore, the final available dose is equivalent to that used for nasal sprays. 

Man et al. [26] provide the only selected study to explore the risk of cataracts being formed. They could not identify any; however, only patients after 6 weeks of treatment were included. 


*
IOP:
*


Two authors explored IOP after short-term steroid irrigation [26,28]; neither found any increase in pressure.


*
HPA axis:
*


Six authors explored the HPA axis after short term treatment [14,15,17,24,25,26]. None of the selected studies could identify any decrease in morning serum cortisol, 24 h urinary cortisol, salivary cortisol, or alteration in the cosyntropin stimulation test after treatment with HSNR.


**
*
Question 7: is HSNR therapy safe in the long term (>3 months)
*
**


There are six studies which evaluate this thesis [13,19,20,23,29,30], including one systematic review and meta-analysis about this topic [19]. None of them could identify any statistically significant increase in IOP or ACTH after treatment. 


*
HPA axis:
*


Four authors explored the HPA axis [19,23,29,30], including one systematic review and meta-analysis [19], and only one of them [30] reported the presence of an alteration. Yoon et al. [19] did not find any increase in ACTH (OR:0.28; 95%CI: 0.04–1.84) in their meta-analysis of four studies.

The work of Soudry et al. [30], who investigated HPAA suppression trough stimulated cortisol testing, is remarkable. They found that 23% of their sample, which received budesonide nasal irrigations for at least 6 months, had developed subclinical adrenal insufficiency. Interestingly, daily budesonide dosage, duration of use, cumulative dose, and patient demographics were not associated with increased risk. However, concomitant use of steroid inhalers was statistically associated with adrenal insufficiency with a 30.4 Odds ratio.


*
High IOP:
*


Three authors explored it [19,23,30], including one systematic review and meta-analysis, and none reported any increase in IOP.

## 4. Discussion

This is the first state-of-the-art review published to date aiming to summarize and critically assess the compendium of available evidence regarding the role of HSNR therapy on CRS in order to guide clinicians in their evidence-based daily practice.

This state-of-the-art review reveals the increasing interest in the use of HSNR for CRS, as most selected studies have been published in recent years. 

There have been previous reviews about this topic [10,15,39] which have been summarized and included hereunder. However, these reviews were either narrative reviews which were not following a systematic approach or else they did not differentiate between CRSwNP and CRSsNP.

The most prominent study was performed by Grayson and Harvey [10]. Contrary to our approach, they only included randomized-controlled trials (RCT) and systematic reviews and; therefore, only 11 studies were selected, while we have included 23. Furthermore, we have differentiated between CRSwNP and CRSsNP in our analysis.

The available evidence and recommendations are summarized in Table 1. Individual data from the selected studies are summarized in Appendix A.


**
Confounding factors
**


Our capacity for obtaining conclusions from the available evidence is highly influenced by the control of confounding factors.


*
Extension of the surgery:
*


It has been reported that the extension of the surgery affects its recurrence rate. Therefore, in order to compare patients in the same study and among studies, the extension of surgery should be standardized. Up to date, there is no standard of care, and this is an important limitation of this study.


*
Benefits of saline irrigation:
*


Irrigations have been shown to increase mucociliary transport and remove mucus, debris, biofilms, and environmental pathogens. Therefore, their effects may have overcome any added positive benefit of budesonide. Thus, the steroid group should always be compared against a group with high-volume saline nasal rinse to elucidate if there is added value in the steroid. Only six studies have been controlled with saline irrigation [15,16,19,21,22,23], including one systematic review and meta-analysis [19].

Yoon et al. [19] could not identify any significant difference comparing the experimental group with the saline rinse controls in their meta-analysis. These results question the usefulness of HSNR in CRS. However, as previously discussed, the design of some of the studies was not appropriate in order to find differences, either due to sample selection, follow-up period, or amount of irrigation. 


*
Mixing CRSwNP and CRSsNP:
*


Ten authors have mixed both diagnoses [6,13,14,16,17,18,20,26,29,30]. Despite both CRSwNP and CRSsNP being called chronic rhinosinusitis, they are different entities. The available evidence, despite the fact that it is scarce, suggests better results in CRSwNP patients.

Luz-Matsumoto et al. [13] included both types of CRS patients. They performed subgroup analysis and found improvements in Lund–Kennedy and SNOT-22 scores only for the CRSwNP subgroup.

Despite not being included in this review because they mixed different nasal lavage protocols, Snidvongs et al. [12] also analyzed their data by subgroups. They discovered differences in NPS and SNOT-22, reporting a higher decrease for CRSwNP.

Tait et al. [16] also carried out an analysis of subgroups and could not ascertain the type of CRS to be a confounding factor in the response to nasal lavage, despite the mean decrease in SNOT-22 being noticeably different (10.2 in CRSwNP and 4.1 in CRSsNP).


*
Previous surgery:
*


Five authors report on patients with and without previous ESS [13,14,16,27,31]. Among them, only three performed subgroup analysis. The evidence obtained is contradictory, so it is not clear whether previous ESS improves the results of HSNR or not.

Firstly, Luz-Matsumoto et al. [13] compared patients with and without previous ESS. They reported that in the group which underwent surgery, only intranasal irrigation (but not sprays) improved the Lund–Kennedy score and had a higher rate of SNOT-22 improvement. They concluded that previous ESS was the best scenario for HSNR use.

Secondly, Talat et al. [31] reported that 78.6% of their sample had undergone previous surgery. After performing a multivariate analysis, they did not find this fact to be related to the final decrease in SNOT-22 nor NPS. 

Finally, and interestingly, Tait et al. [16] mixed patients with and without previous ESS. They performed a subgroup analysis and reported a noticeable difference in the SNOT-22 mean decrease (0.1 if previous ESS; 10.1 if no prior ESS). They could not give any explanation for their findings. However, it could be explained by arguing that the initial SNOT-22 values could have been different between groups (lower because of the previous ESS; or higher as ESS was performed only in severe cases). 


*
CRS endotypes
*


Despite some authors having classified their patients according to phenotype into CRSwNP and CRSsNP, none have classified them according to endotype. There is increasing evidence that the endotype rather than the phenotype is the most appropriate guide for treatment choice [40]. Different CRS endophenotypes would be the most appropriate way to compare outcomes. We encourage future studies to follow this approach.

## 5. Conclusions

The available evidence suggests a potential positive effect of HSNR, which seems to be higher in CRSwNP. More well-designed studies are needed in order to obtain firm conclusions. Nevertheless, the evidence is solid regarding the safety of this treatment in the short and long-term. We expect that the lack of severe negative effects will facilitate the acceptance of this treatment modality and the development of future studies.

## Figures and Tables

**Figure 1 jcm-12-03605-f001:**
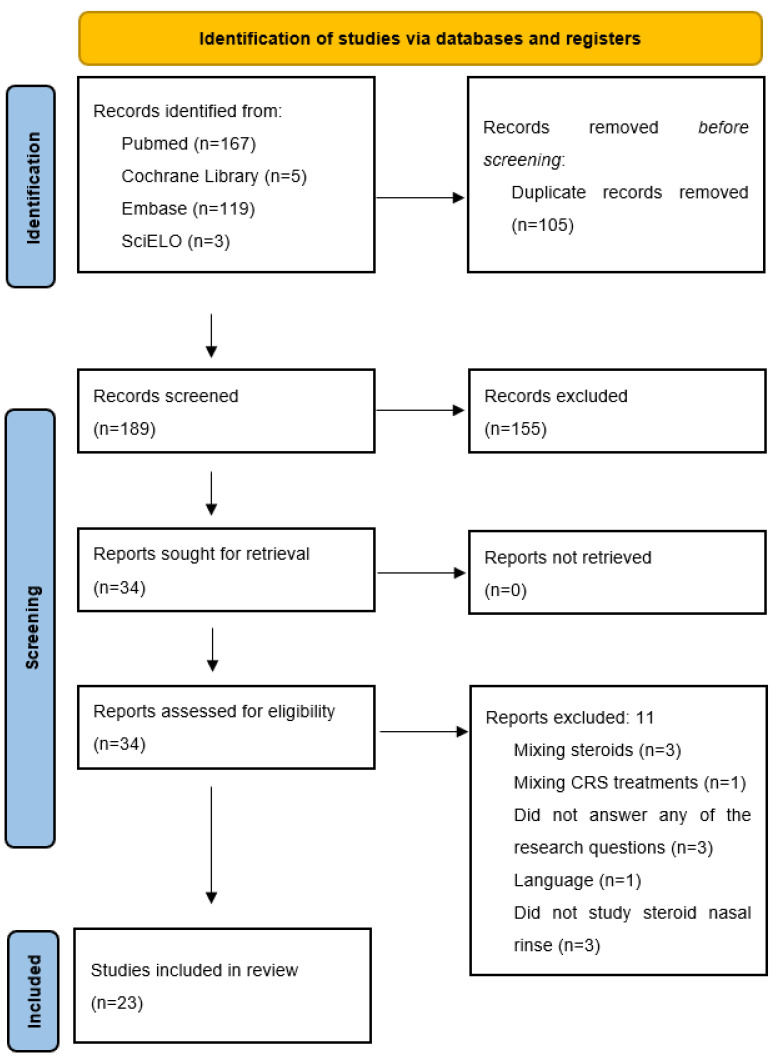
Prisma flow-diagram.

**Table 1 jcm-12-03605-t001:** Summary of the available evidence. References in superscript number.

**Question 1a: is steroid diluted high-volume nasal rinse therapy useful in CRS without ESS?—Mixed CRS**	
**Available evidence**	None
**Level of evidence**	Not applicable
**Conclusions**	There is no available evidence on this topic
**Question 1b: is steroid diluted high-volume nasal rinse therapy useful in CRS without ESS?—CRSwNP**	
**Available evidence**	None
**Level of evidence**	Not applicable
**Conclusions**	There is no available evidence on this topic
**Question 1c: is steroid diluted high-volume nasal rinse therapy useful in CRS without ESS?—CRSsNP**	
**Available evidence**	Favoring: 1 RCT [15]
**Level of evidence**	Level 2
**Conclusions**	HSNR may decrease nasal symptoms in CRS without ESS. More studies are needed.
**Question 2a: is steroid diluted high-volume nasal rinse therapy useful in CRS after ESS?—Mixed CRS**	
**Available evidence**	Favoring: 2 RCT [16]; 2 Quasi-experimental [6,17]; 1 Cohort study [18] Against: 1 Systematic review [19]; 1 RCT [20]
**Level of evidence**	Level 1
**Conclusions**	HSNR may not decrease nasal symptoms in mixed CRS. More studies with well selected controls are needed.
**Question 2b: is steroid diluted high-volume nasal rinse therapy useful in CRS after ESS?—CRSwNP**	
**Available evidence**	Favoring: 1 Quasi-experimental [21] Against: 2 RCT [22,23]
**Level of evidence**	Level 2
**Conclusions**	The evidence is contradictory. More studies are needed to solve this question.
**Question 2c: is steroid diluted high-volume nasal rinse therapy useful in CRS after ESS?—CRSsNP**	
**Available evidence**	Against: 1 RCT [24]
**Level of evidence**	Level 3
**Conclusions**	The available evidence is of insufficient quality to answer this question.
**Question 3: which treatment is better?**	
**Available evidence**	1 RCT [13]
**Level of evidence**	Level 2
**Conclusions**	There is scarce evidence. Available evidence suggests that higher doses offer better symptom control. Budesonide seems to be more effective than betamethasone. High-volume rinse seems to offer better symptom control than low-volume rinses.
**Question 4: is steroid diluted high-volume nasal rinse better than intranasal steroid spray?**	
**Available evidence**	Favoring: 1 Systematic review [10]; 4 RCT [13,15,20,23]; 1 Cohort study [18]
**Level of evidence**	Level 1
**Conclusions**	All the available evidence suggests that HSNR offers better symptom control than INCS
**Question 5: which patients are most benefited from steroid nasal rinses?**	
**Available evidence**	1 Quasi-experimental [6]; 1 Cohort [18]
**Level of evidence**	Level 3
**Conclusions**	To date, there is no evidence of which patients are more benefited from HSNR. AERD patients may have worse results than non-AERD patients.
**Question 6: is steroid diluted high-volume nasal rinse therapy safe in the short term (<3 months)?**	
**Available evidence**	Favoring: 5 Quasi-experimental study [14,17,25,26,27,28]; 2 RCT [15,24]
**Level of evidence**	Level 2
**Conclusions**	Available evidence is strong. There is no evidence of complications in the short term of HSNR
**Question 7: is steroid diluted high-volume nasal rinse therapy safe in the long term (>3 months)**	
**Available evidence**	Favoring: 1 Systematic review [19]; 2 Cross-sectional study [29]; 1 Cohort study [13]; 2 RC T [20,23] Against: 1 Cross-sectional study [30]
**Level of evidence**	Level 1
**Conclusions**	Available evidence is strong. There is no evidence of complications in the long term of HSNR

**Table 2 jcm-12-03605-t002:** Assessment of the risk of bias quasi-experimental and cohort study: (1.1) Is the source population or source area well described? (1.2) Is the eligible population or area representative of the source population or area? (1.3) Do the selected participants or areas represent the eligible population or area? (2.2) Were interventions well described and appropriate? (2.4) Were participants or investigators blind to exposure and comparison? (2.5) Was the exposure to the intervention and comparison adequate? (2.8) Were all participants accounted for at study conclusion? (2.9) Did the setting reflect usual practice? (3.1) Were outcome measures reliable? (3.2) Were all outcome measurements complete? (3.3) Were all important outcomes assessed? (3.4) Were outcomes relevant? (4.1) Were exposure and comparison groups similar at baseline? (4.2) If not, were these adjusted? (4.3) Was the study sufficiently powered to detect an intervention effect (if one exists)? (4.4) Were the estimates of effect size given or calculable? (4.5) Were the analytical methods appropriate? (4.6) Was the precision of intervention effects given or calculable? (5.1) Are the study results internally valid (i.e., unbiased)? (5.2) Are the findings generalizable to the source population (i.e., externally valid)?

	Kosugi (2016) [6]	Sachanandani (2009) [14]	Dawson (2017) [17]	Kang (2017) [21]	Welch (2010) [25]	Man (2013) [26]	Bhalla (2008) [27]	Seiberling (2013) [28]
1.1	+	+	+	−	++	++	−	+
1.2	+	+	+	+	+	++	−	+
1.3	+	−	++	−	++	++	−	++
2.2	++	++	++	++	++	++	+	++
2.4	−	−	NA	−	−	−	−	−
2.5	+	++	+	+	++	++	++	+
2.8	++	++	++	++	−	+	++	++
2.9	++	+	++	++	++	++	+	+
3.1	+	+	++	+	+	+	+	+
3.2	++	+	++	++	−	++	++	+
3.3	+	++	+	+	++	++	++	++
3.4	++	++	−	++	++	++	++	++
4.1	++	++	++	++	+	++	+	+
4.3	−	−	+	−	−	−	−	−
4.4	−	−	−	−	−	−	−	−
4.5	+	+	+	+	+	++	+	+
4.6	+	+	+	+	+	+	+	+
5.1	+	−	+	+	−	+	+	+
5.2	+	−	++	+	−	++	−	+

NA = not applicable; ++ = well covered; + = adequately addressed; − = poorly addressed.

**Table 3 jcm-12-03605-t003:** Assessment of the risk of bias for clinical trial. (A1) An appropriate method of randomization was used to allocate participants to treatment groups (which would have balanced any confounding factors equally across groups); (A2) There was adequate concealment of allocation (such that investigators, clinicians and participants cannot influence enrolment or treatment allocation); (A3) The groups were comparable at baseline, including all major confounding and prognostic factors; (B1) The comparison groups received the same care apart from the intervention(s) studied; B2 Participants receiving care were kept ‘blind’ to treatment allocation; (B3) Individuals administering care were kept ‘blind’ to treatment allocation; (C1) All groups were followed up for an equal length of time (or analysis was adjusted to allow for differences in length of follow-up); (C2) The groups were comparable for treatment completion (that is, there were no important or systematic differences between groups in terms of those who did not complete treatment); (C3) The groups were comparable with respect to the availability of outcome data (that is, there were no important or systematic differences between groups in terms of those for whom outcome data were not available); (D1) The study had an appropriate length of follow-up; (D2) The study used a precise definition of outcome; (D3) A valid and reliable method was used to determine the outcome; (D4) Investigators were kept ‘blind’ to participants’ exposure to the intervention; (D5) Investigators were kept ‘blind’ to other important confounding and prognostic factors.

	Luz-Mats (2021) [13]	Jiramongkolchai (2020) [15]	Tait, S. (2018) [16]	Harvey (2018) [20]	Rawal (2015) [22]	Rotenberg (2011) [23]	Thamboo (2014) [24]	Talat, R. (2021) [31]
A1	−	+	++	++	++	NR	+	−
A2	−	+	++	++	++	NR	+	+
A3	+	+	+	++	++	+	+	NA
B1	+	++	++	++	++	++	++	NA
B2	NA	++	++	++	++	++	−	NA
B3	NA	++	++	++	++	++	−	NA
C1	+	++	++	++	++	++	++	++
C2	−	+	++	++	++	++	++	++
C3	++	+	++	++	++	++	++	+
D1	+	+	+	++	+	++	+	+
D2	++	++	++	++	++	++	++	+
D3	++	++	++	++	++	++	++	+
D4	NA	+	+	+	++	++	+	−
D5	NA	+	NR	+	NR	+	NR	−

NA = not applicable; NR = not reported; ++ = well covered; + = adequately addressed; − = poorly addressed.

## Data Availability

All the available data is published in this paper.

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
