# Peer review of "The Role of Corticosteroid Nasal Irrigations in the Management of Chronic Rhinosinusitis: A State-of-the-Art Systematic Review"

_jcm, 2023, doi:10.3390/jcm12103605_

Round 1
Reviewer 1 Report
This is a systematic review on the role of corticosteroid nasal irrigations in CRS irrigations, that eventually analyzed 23 studies, including 722 cases and 460 controls patients. I congratulate the authors by this throughout review on this topic, as nasal steroids are still the mainstay therapy for CRS patients. Despite this subject has already been explored by other papers in different forms, it is important to update the current body of evidence with more robust methodologies, such as used in this study. Overall, the paper is well written and objectively brings a summary of answers to relevant questions about corticosteroid nasal irrigation in CRS patients.
I have few minor comments/questions about this paper:
1- I suggest including “nasal polyps” in the keywords.
2- In the introduction (and is extended to the discussion), I fell that the authors could explore some limitations on the current evidence that all most treatments compared the outcomes based only on the phenotypic presentation of CRS (polyp vs non-polyp disease). This classification is known to be not sufficient to predict the underlying pathoghenesis. My suggestion is to include some comments that different CRS endophenotypes would be the most appropriate way to compared outcomes, but there are still little evidence available at this point in time.
3- In “Material and Methods”, please exclude lines 67-81 (probably these are the journal`s orientations to the authors). The same apply to lines 162-164 in discussion.
4- I believe that most data presented in “discussion” should be in section of “results” (from lines 179 to 358). I would reserve the space of discussion to comment on the highlights of the study, strengths, and limitations of the study or of current evidence. The subsection “confounding factor”, “extension of surgery”, “benefits of saline irrigation”, “previous surgery”, and “CRS endotypes”, for instance, I would leave in discussion.
5- For a better understanding, in line 187, I suggest changing “Mixed CRSwNP and CRSsNP” by “Mixed CRS (CRSwNP and CRSsNP).”
Author Response
This is a systematic review on the role of corticosteroid nasal irrigations in CRS irrigations, that eventually analyzed 23 studies, including 722 cases and 460 controls patients. I congratulate the authors by this throughout review on this topic, as nasal steroids are still the mainstay therapy for CRS patients. Despite this subject has already been explored by other papers in different forms, it is important to update the current body of evidence with more robust methodologies, such as used in this study. Overall, the paper is well written and objectively brings a summary of answers to relevant questions about corticosteroid nasal irrigation in CRS patients.
Thank you for your time and effort reviewing our manuscript.
I have few minor comments/questions about this paper:
- I suggest including “nasal polyps” in the keywords.
It has been included
- In the introduction (and is extended to the discussion), I fell that the authors could explore some limitations on the current evidence that all most treatments compared the outcomes based only on the phenotypic presentation of CRS (polyp vs non-polyp disease). This classification is known to be not sufficient to predict the underlying pathoghenesis. My suggestion is to include some comments that different CRS endophenotypes would be the most appropriate way to compared outcomes, but there are still little evidence available at this point in time.
We agree. In fact, it was already included in the discussion (line 490-494 of the former document). This point has been extended (line 466-469)
- In “Material and Methods”, please exclude lines 67-81 (probably these are the journal`s orientations to the authors). The same apply to lines 162-164 in discussion.
Thank you, these paragraphs have been eliminated.
4- I believe that most data presented in “discussion” should be in section of “results” (from lines 179 to 358). I would reserve the space of discussion to comment on the highlights of the study, strengths, and limitations of the study or of current evidence. The subsection “confounding factor”, “extension of surgery”, “benefits of saline irrigation”, “previous surgery”, and “CRS endotypes”, for instance, I would leave in discussion.
Ok, in this kind of studies is difficult to distinguish clearly what is a result and what is a conclusion of these results. We have moved it to the results section.
- For a better understanding, in line 187, I suggest changing “Mixed CRSwNP and CRSsNP” by “Mixed CRS (CRSwNP and CRSsNP).”
Done (now line 202)
Reviewer 2 Report
Thank you very much for considering me as a reviewer for the manuscript: “The role of Corticosteroid nasal irrigations in the management of chronic rhinosinusitis. A state-of-the-art review” by Christian Calvo-Henriquez et al. I was pleased to agree to review the work because despite the awareness that the use of this type of therapy is "out of label" it is commonly used in CRS treatment. Personally, I use budesonide high volume solution in my practice and that's why I was interested in the results of the review. Among many unanswered questions of this treatment the most important is how to reach effective steroid solution? what should be appropriate concentration of the steroid in such a solution? In some cases, we may afraid that it is homeopathic solution. Another question is safety of the use of the steroids in nasal lavage and optimal period of treatment? Based on review authors did not give the response for these questions suggesting that there is room for many research studies which will give the answer. I’m aware of imperfection of the studies which are base for review but I would recommend suggestive summary of the tendency in which the high-volume steroid solution therapy study should be developed.
Correct line 258 : “ministration” to administration
Summarizing this is interesting review containing currently available knowledge in this field and is a basis for further research to create study to reach the effectiveness of security and target group of that treatment.
Author Response
Thank you very much for considering me as a reviewer for the manuscript: “The role of Corticosteroid nasal irrigations in the management of chronic rhinosinusitis. A state-of-the-art review” by Christian Calvo-Henriquez et al. I was pleased to agree to review the work because despite the awareness that the use of this type of therapy is "out of label" it is commonly used in CRS treatment. Personally, I use budesonide high volume solution in my practice and that's why I was interested in the results of the review. Among many unanswered questions of this treatment the most important is how to reach effective steroid solution? what should be appropriate concentration of the steroid in such a solution? In some cases, we may afraid that it is homeopathic solution. Another question is safety of the use of the steroids in nasal lavage and optimal period of treatment? Based on review authors did not give the response for these questions suggesting that there is room for many research studies which will give the answer. I’m aware of imperfection of the studies which are base for review but I would recommend suggestive summary of the tendency in which the high-volume steroid solution therapy study should be developed.
Thank you for your time spent reviewing our manuscript. As you, we hope that this paper may guide other authors (if not us) to answer some of the currently unanswered questions as you have pointed out.
Correct line 258 : “ministration” to administration
Corrected (now line 279)
Summarizing this is interesting review containing currently available knowledge in this field and is a basis for further research to create study to reach the effectiveness of security and target group of that treatment.
Thank you